# Taking a BiTE out of Lymphoma: Bispecific Antibodies in B-Cell Non-Hodgkin Lymphoma

**DOI:** 10.3390/cancers16091724

**Published:** 2024-04-28

**Authors:** Jonathan M. Weiss, Tycel J. Phillips

**Affiliations:** 1Rogel Comprehensive Cancer Center, University of Michigan, Ann Arbor, MI 48109, USA; weissjon@med.umich.edu; 2City of Hope Comprehensive Cancer Center, Department of Hematology and Hematopoietic Cell Transplantation, Division of Lymphoma, Duarte, CA 91010, USA

**Keywords:** NHL, BsABs, bispecifics, DLBCL, FL, MCL, treatment, management

## Abstract

**Simple Summary:**

Bispecific antibodies are a novel type of immune-based therapy used to treat patients with non-Hodgkin’s lymphoma. This review article will help define the current treatment landscape of non-Hodgkin’s lymphoma and detail ways in which bispecific antibodies fit in the current management paradigm, with special attention paid to diffuse large B-cell lymphoma, follicular lymphoma, and mantle cell lymphoma.

**Abstract:**

B-cell non-Hodgkin’s lymphoma (NHL) refers to a heterogenous group of diseases, all of which have a wide range of treatment strategies and patient outcomes. There have been multiple novel, immune-based therapies approved in NHL in the last decade, including bispecific antibodies (BsAbs) and chimeric antigen receptor therapy (CAR-T). With a host of new therapies, an important next step will be determining how these therapies should be sequenced in contemporary management strategies. This review seeks to offer a framework for the ways in which BsABs can be incorporated into the current management paradigm for NHL, with special attention paid to diffuse large B-cell lymphoma (DLBCL), follicular lymphoma (FL), and mantle cell lymphoma (MCL).

## 1. Introduction

Non-Hodgkin’s lymphoma (NHL) refers to a diverse group of malignant neoplasms that arise from clonally proliferative B cells, T cells, and NK cells. B-cell NHL is the most common subtype within the US, with diffuse large B-cell (DLBCL) and follicular lymphoma (FL) accounting for most of these cases. In the last decade, there have been multiple new United States (US) Food and Drug Administration (FDA) therapeutic approvals in NHL. These novel therapeutics have included intracellular [1,2,3] and extracellular targeting agents [4,5,6], as well as therapies that activate an endogenous anti-tumor immune response [7,8,9,10,11,12]. Among these novel therapies are the bispecific antibodies (BsAbs), which bind two specific antigens simultaneously and redirect T cells or Natural Killer cells to tumor-associated antigens. Recently, several BsAbs have been approved by the FDA for relapsed/refractory (R/R) DLBCL and FL [7,11,13], signaling a time of optimism for the on-going management of patients with R/R NHL. However, with multiple new therapies, new questions arise as to how we should sequence these therapies into both the front-line (1 L) and R/R setting. The goal of this review article is to discuss these novel BsAbs, provide insight into how we can consider sequencing BsAbs, and take a look at what might be coming in the near future.

## 2. DLBCL and BsAbs

DLBCL is the most common type of lymphoma in the United States of America and accounts for approximately 30% of all cases of NHL and more than 20,000 annual cases [14]. Front-line therapy can cure approximately 60–70 percent of patients [15,16], leaving up to 40% of patients with inadequate responses to initial therapy. These patients, especially those who are refractory to 1 L therapy, have poor outcomes and are in need of novel treatments. A major advance has been the approval of chimeric antigen receptor therapy (CAR-T), which has dramatically improved outcomes in this space, but access to this treatment is limited [17]. With a similar mechanism of action, the BsAbs offer another promising alternative with improved accessibility with relation to CAR-T. Two BsAbs have recently been approved by the US FDA for the management of R/R DLBCL, epcoritamab and glofitamab, based on the results of two pivotal phase 2 trials [7,11,12]. Additionally, Odronextamab, while not approved at this time in r/r DLBCL, has shown encouraging clinical activity as well [18]. 

Epcoritamab is an IgG full-length monoclonal anti-CD20/CD3 Bispecific Antibody (BsAb). It was evaluated in a phase 1/2 study of patients with R/R CD20+ large B-cell lymphoma with at least two prior lines of therapy [12]. It is administered subcutaneously in 28-day cycles with step-up dosing (SUD) (C1D1: 0.16 mg → C1D8: 0.8 mg → C1D15: 48 mg → C1D21 and beyond: 48 mg). On cycles 1–3, epcoritamab is given on days 1, 8, 15, and 21. During cycles 4–9, it is administered on days 1 and 15. From cycles 10 and beyond, it is administered on day 1. The drug is then continued until disease progression or intolerance. The overall response rate (ORR) in the R/R DLBCL patient population was 63.1% (99/157), with a complete response rate (CRR) of 38.9% (61/157) and a median duration of response (DOR) of 12 months. Adverse events include cytokine release syndrome (CRS), which occurs in 52% of patients (Grade 1: 34%, Grade 2: 15%, Grade 3: 3%). Most cases of CRS occur 20 h after administration on C1D15, and hospitalization is recommended at C1D15. CRS mitigation strategies include acetaminophen, diphenhydramine, and corticosteroids prior to cycle 1 treatments (corticosteroids should also be continued for three days following each cycle 1 treatment) [19]. In a recent update, study investigators have explored an “optimization strategy”, which included the use of intravenous (IV) fluids and, specifically, dexamethasone for a corticosteroid. Patients were encouraged to drink 2–3 L of fluid the day before starting therapy; patients would receive 500 L of IV fluid on the day of the injection, followed by another 2–3 L of fluid the day after the injection. With this strategy, they noted an approximately 50% reduction in CRS events [20].

Glofitamab is also approved for R/R DLBCL. Glofitamab is a humanized mouse-derived bispecific antibody with 1:2 (CD3:CD20 ratio) Fab arms. It was evaluated in a phase 1/2 study in patients with r/r DLBCL who have received at least two or more prior lines of therapy [7]. Glofitamab is given in 21-day cycles after completion of SUD. Unique to this agent, it is started 7 days after a pre-phase dose of 1000 mg of obinutuzumab. This is believed to reduce the risk of CRS due to decreasing the amount of circulating B cells prior to the receipt of glofitamab. Thereafter, glofitamab is given at a dose of 2.5 mg on day 8, then 10 mg on day 15, and the full dose of 30 mg is then administered on day 1 of cycle 2. It is then continued at this dose every 3 weeks through cycle 12. The study reported an ORR of 52% (80/155) and a CRR of 39% (61/155), and the median DOR was 18.4 months. Recently, updated data noted that patients who obtained a CR had a DOR of 26.9 months and a PFS of 31.1 months, with approximately 55% of these patients still in remission at 24 months [21]. Patients remained on treatment for 12 months (or until progression or intolerance), in contrast to epcoritamab, in which patients remained on treatment indefinitely if responding and tolerating therapy. The most common adverse event, as expected for the class, was CRS. CRS occurred in 63% (97/154) of patients and most commonly occurred following C1D8 (2.5 mg dose), at about 13.5 h after administration (range 6–52 h), with a median duration of CRS of 30.5 h. To mitigate CRS, in addition to pre-treatment obinutuzumab, patients received pre-dose acetaminophen, diphenhydramine, and dexamethasone 20 mg prior to all doses of glofitamab. While not originally in the study, a further evaluation did suggest that the addition of dexamethasone significantly reduced the risk of CRS during SUD [22]. 

Odronextamab, while not yet approved, has shown clinical activity in patients with r/r DLBCL [18]. Odronextamab is a hinge-stabilized, human IgG4 CD20 X CD3 BsAB. In an effort to mitigate CRS, the regimen is given in 21-day cycles with SUD and split days. In cycle 1, the regimen consists of 0.7 mg split over D1/D2, 4 mg split over D8/D9, and 20 mg over D15/D16. For cycles 2–4, 160 mg is given on D1, D8, and D15. After cycle 4, maintenance treatment continues at 320 mg every 2 weeks until disease progression or unacceptable toxicities. The ORR and CRR were 52% (66/127) and 31% (38/127), respectively. The median DOR was 10.2 months, with a median duration of CR of 17.9 months. CRS was the most common adverse event, and it occurred in 55% of patients. Only one grade 3 CRS event occurred in the setting of pancreatitis. Overall, odronextamab has highly favorable clinical activity, with a subset of patients obtaining deep and durable responses. The split and SUD may help mitigate CRS, as seen by the low rates of grade 3 CRS. However, when it comes to clinical practice, epcoritamab or glofitamab may be a more feasible/convenient option. Odronextamab requires multiple clinic visits for drug administration. It also takes 9 weeks until the full dose is given, which may impact response pending aggressiveness of the disease.

Despite the inherent benefit of BsAbs relative to CAR-T, especially in community and/or non-CAR centers, there has been a lack of robust uptake. The major limitation to the integration of BsAbs into these settings remains the management of CRS. Given the limited experience of community providers with CRS and its management, a learning curve is to be expected. Even within the academic centers, strategies for CRS management vary from site to site. Therapeutic strategies for managing CRS include the administration of acetaminophen, dexamethasone, and tocilizumab, along with hospital admission. Recently a detailed multi-institutional guideline for the management of CRS was published which provides a guideline for physicians and institutions to follow [19]. Even though most patients will not need tocilizumab, it remains a barrier for sites given it is a cost-prohibitive therapy with a limited shelf-life, so further thought to overcome this is needed. Given that most CRS events are limited to SUD, a strategy for eliminating the burden of hospitalization is partnering with academic sites until further data support fully giving these drugs as an outpatient. 

In patients with DLBCL that relapse within one year of front-line therapy, or are refractory to front-line therapy, the standard of care second-line treatment is CAR-T with both axicabtagene ciloleucel (Axi-Cel) or lisocabtagene maraleucel (Liso-Cel), currently having FDA approval [23,24]. For this patient population, there is not a sequencing issue currently, as we have no firm data on the efficacy of BsAbs in the second-line setting (2 L). There are several studies that have evaluated BsAbs in combination with other agents. These have included chemotherapy, as well as targeted agents (Epco-GemOx, Epco-DHAOx, Epco-Len, Mosun-Pola, and Glofit-Gemox). While these trials are being studied in a patient population that would not generally be considered for CAR-T in 2 L, it still gives a hint of the efficacy of BsAbs in 2 L. This is important given that the data suggest that at least 50% of patients eligible for CAR-T actually receive CAR-T. We also know that in patients with a high burden of disease, the manufacturing time for CAR-T cell therapy, which can take on the order of weeks, may be too long to wait for a novel immune-based therapy. Other unique situations exist, such as when patients’ socioeconomic status, distance from a CAR center, or lack of familial support may prevent them from receiving CAR-T. This later point is evident in the disparities in access to commercial and clinical trial CAR-T cell therapy, especially among African Americans, Hispanic populations, and those from neighborhoods with low median income [17]. In these patients, it may be beneficial to proceed with a BsAb therapy, which is “off the shelf”, and, as such, does not require prolonged manufacturing times, can be given locally, and does not require a patient to live close to a CAR-T center for a prolonged period. In patients who are receiving third-line or later treatment (3 L+), the maturity of the data with CAR-T still supports the utilization of these agents prior to BsAbs unless a situation arises which limits the accessibility of the CAR product. As compared to other FDA-approved agents for 3 L+ disease, the BsAbs compare favorably and should be considered as a first option. Additionally given the functionality and toxicity profile, the BsAbs, as mentioned, are well positioned to be combined with other drugs to improve efficacy. 

## 3. FL and BsABs

FL is the second most common lymphoma worldwide and accounts for approximately 17% of patients with NHL, with an annual incidence rate of 3.23 per 100,000 people [14,25]. In extensive-stage disease, the goal of therapy is disease control and palliation of symptoms, as current front-line therapies do not offer cure. The standard of care front-line therapies for medically fit patients generally includes a CD20 monoclonal antibody with either chemoimmunotherapy (bendamustine, CVP, or CHOP) or lenalidomide [26,27]. In patients with R/R FL, several options exist, with lenalidomide being one the most widely used 2 L options, as well as chemoimmunotherapy. For those with 3 L+ disease, both CAR-T cell therapy or BsAB have been approved [13,28,29,30]. Currently, mosunetuzumab is the only BsAb approved for patients with R/R FL, while both odronextamab and epcoritamab have shown efficacy in clinical trials recently [31,32].

Mosunetuzumab is a full-length, IgG1-based CD20 × CD3 bispecific monoclonal antibody that is currently approved for patients with R/R FL who have received two or more prior lines of therapy [13]. It is given intravenously in 21-day cycles after completion of SUD. In cycle 1, patients undergo SUD (C1D1: 1 mg → C1D8: 2 mg → C1D15 and C2D1: 60 mg → and then 30 mg every 21 days until either cycle 8 or 17 depending on attainment of a complete response). The ORR and CRR in R/R FL were 80% (72/90) and 54/90 (60%), respectively. The median DOR was 22.8 months. The most common adverse event was CRS, which occurred in 44% (40/90) of patients, and was Grade 3–4 in approximately 2% (2/90) of patients. CRS occurred following C1D15 in 36.4% of cases, with a median time to CRS onset of 27 h, and a median duration of CRS of 3 days. To mitigate CRS, acetaminophen, diphenhydramine, and dexamethasone are given prior to each dose in cycle 1 and cycle 2. Importantly, of the approved BsAbs, mosunetuzumab is the only one without a recommendation for hospitalization after any dose during SUD. This allows for more flexibility for the treating physician when determining how to best observe CRS in these patients.

Both odronextamab and epcoritamab are likely the next BsAbs to be approved in FL [31,32]. Odronextamab was studied in 121 patients with r/r FL and followed a SUD with split-day therapeutic plan, similar to what was described in r/r DLBCL. They reported an ORR of 81.8%, CRR of 75.2%, and mDOR of 20.5 months, with 55.3% of patients still in remission at 18 months [32]. Epcoritamab reported an ORR of 82% and a CRR 63% with still-maturing data with respect to DOR and PFS [31]. When these therapies will get an indication is still to be determined, as well as how they will compete with mosunetuzumab. Mosunetuzumab seems like the auspicious choice when compared to odronextamab and epcoritamab, given that (1) it has a more favorable SUD regimen than odronextamab; (2) it has a finite treatment duration, while both odronextamab and epcoritamab are given until progression; and (3) epcoritamab currently has a recommendation for hospitalization.

With respect to sequencing, there have been no clinical trials that have compared BsAbs to CAR-T cell therapy. In FL, there are also currently no data that indicate that CAR-T is curative. Providers should take a patient-centric approach with respect to these treatments. For example, in patients in which there is concern for transformation from FL to DLBCL (i.e., “hot” FDG avid lymph nodes, high-risk clinical features, etc.), CAR-T cell therapy may be the preferred option, as it has shown curative potential in DLBCL. However outside of this scenario, mosunetuzumab may be a more favorable option. This is due to the ability for mosunetuzumab to be given in a variety of settings, as it does not require hospitalization and can be administered safely in a community setting. The toxicity profile for mosunetuzumab also compares quite favorably to CAR-T, given the almost complete lack of neurological complications, lower and more manageable rates of CRS, and lower risk of prolonged cytopenias and infection.

## 4. Mantle Cell Lymphoma (MCL) and BsABs

MCL is another B-cell NHL in which BsABs have been evaluated. MCL can be clinically variable in presentation with both indolent and highly proliferative/aggressive presentations. Advanced-stage MCL is not considered curable with contemporary management strategies, and novel therapies are needed for patients with r/r disease. Front-line management of MCL varies depending on multiple factors (e.g., biology/molecular characteristics, patient functional status, etc.) but generally consists of chemoimmunotherapy (CIT) ± autologous stem cell transplant. Additionally, Bruton Tyrosine Kinase (BTK) BTK inhibitors have become an acceptable option in either front-line or r/r MCL. The evaluation of BsABs in MCL has lagged behind both DLBCL and FL thus far. This is in part related to the rarity of the disease and the difficulty thus far with safely administering the drug in MCL patients. While several BsABs have included MCL cohorts in their initial studies, most have not published or presented any data. Today, the only BsABs with available data are mosunetuzumab and glofitamab. Glofitamab was evaluated in patients with MCL who relapsed post-BTK inhibitor in phase I/II trial with 37 patients. The ORR and CRR were 83.8% (31/37) and 73.0% (27/37), respectively. As expected, CRS was the most common adverse event, occurring in 75.7% of patients, with only two grade 4 CRS events. Immune effector cell-associated neurotoxicity (ICANS) occurred in five patients, and was grade 1/2. Single-agent mosunetuzumab was evaluated in a phase 1/1b multicenter expansion study in patients with r/r NHL, which included patients with MCL [33]. The ORR and CRR were relatively unimpressive at 4/13 (30.8%) and 3/13 (23.1%), respectively. Notably, 6/13 (46.2%) patients had progressive disease. While there likely is not a future for single-agent mosunetuzumab, the combination of mosunetuzumab and polatuzumab vedotin (M-Pola) has been evaluated in a phase Ib/II study in patients with MCL and progressive disease post BTK inhibitor [34]. Of the 20 patients who received M-Pola, the ORR and CRR were 75% and 70% respectively. CRS occurred in 45% of patients, all of which were grade 1–2. ICANS was less frequent, occurring in only 3/20 (15%) of patients, all of which were grade 1–2. 

Similarly to the situation in DLBCL and FL, how to integrate BsABs into the community at large remains a problem to be solved. Unlike FL, where hospitalization is not required with the approval of mosunetuzumab, single-agent therapy with BsABs in MCL is likely to require an inpatient safety observation period. This is due in no small part to the higher rates of CRS seen in MCL as compared to FL and even DLBCL. In the trial evaluating glofitamab in patients with R/R MCL, the rate of CRS was 76%, with the majority (17 out of the 28 events) being grade 2 or above, suggesting that most will be hospitalized at some point during SUD. Thankfully, as we have seen in 1 L DLBCL and in the combination study of Mosun/Pola in R/R MCL, it appears that combining BsABs with agents with cytotoxic activity might mitigate some of this concern. As stated above, the latter trial was noted to have a CRS rate of 45%, but most (88%) were grade 1. This suggests that, potentially, a way to improve access to the community at large is to use BsABs in combination even if only during SUD to mitigate CRS. Another major discussion will be to determine where BsAbs fit in the current treatment schema for MCL. Currently, there are no BsABs approved in r/r MCL. However, we anticipate approval in the near future. CAR-T cell was recently approved in r/r MCL with excellent, durable responses [35]. The way in which to sequence BsABs relative to CAR-T cell in r/r MCL is yet to be determined. Similar to FL, CAR-T is not curative in MCL, so comparing duration of response, toxicity, and cost as compared to BsABs will be important. Specifically, one key area where BsABs have an early advantage over CAR-T is with respect to ICANS. This especially holds true when compared to brexucabtagene and less so with lisocabtagene but needs to be kept in mind given that ICANS is not always reversible. In patients with r/r MCL pending the long-term durability of response, BsAB will likely be an enticing option. Once approved, we anticipate that BsAB will be a reasonable option in patients who progress or relapse post-BTK inhibitor. Similar to both FL and DLBCL, there are ongoing clinical trials evaluating BsAB in earlier lines and in combination with other MCL-approved agents in patients with MCL—stay tuned, as the next few years will likely greatly expand our understanding of BsABs in this patient population. 

## 5. Resistance

While early response rates and durability have been favorable in DLBCL and FL with yet-to-be-published long-term response data in MCL, treatment failure is an issue. Some groups have already published data that show mechanisms of resistance to these therapies. For example, in patients with DLBCL treated with BsABs, patients with specific genetic alterations (i.e., *TP53*, *RHOA*, and *GNAI2*) have an inferior response when compared to patients without those alterations [36]. However, this is similarly noted in other therapies as well, especially those who harbor TP53 alterations. Whether earlier (less CIT-exposed patients) use of BsABs can overcome this will be revealed in some of the upcoming 1 L trials in less aggressive NHL that are avoiding use of CIT. Similarly, the loss of CD20 expression, and CD20 gene mutations were associated with progressive or recurrent disease in patients with B-cell lymphomas treated with BsABs [37]. This will be an important discussion moving forward, especially pending the persistence of the loss of this important receptor, given the number of drugs which are designed to target CD20 or appear to be secondarily dependent on its expression for response. Future research will look to find mechanisms to promote re-expression or alternatively dual targeting of epitopes (CD19 or CD22) to reduce this mechanism of escape. The tumor microenvironment (TME) likely also plays a role in mechanisms of resistance to BsAB. Patients with progressive disease are more likely to have high levels of PD-1 expression on cytotoxic T lymphocytes and a more “immunosuppressive” TME [38]. A better understanding of these mechanisms of resistance will help us improve outcomes in patients treated with BsABs.

## 6. Future Directions

“Upward mobility” is a key phrase for BsAbs. As previously mentioned, BsAbs are drugs that are uniquely positioned to be able to use as single agents or combine safely with other drugs. To that point, clinical trials are currently evaluating the role of BsAbs as single agents or in combinations in untreated patients. In FL, Falchi et al. presented a phase II study at the American Society of Hematology (ASH) 2023 meeting evaluating a single-agent mosunetuzumab in patients with untreated high-tumor-burden FL [39]. The authors reported that the best overall response rate was 96%, and the CR rate was 81%. Additionally, an ongoing study is looking at mosunetuzumab in FL patients with low tumor burden. A trial of epcoritamab in combination with lenalidomide and rituximab in untreated FL reported favorable results, with an ORR of 90% (26/29) and a manageable safety profile [40]. For DLBCL, BsAbs are being explored in combination with CHOP, R-CHOP, and CHP + polatuzumab [41,42]. Glofitamab + R-CHOP was evaluated in a phase 1b study in front-line setting, with the complete metabolic response (CMR) of 83.9% (47/56). Among those with a complete response, the probability of ongoing response at 12 months was 91.5%, with an overall manageable safety profile [42]. Similarly, glofitamab has been added to Pola-R-CHOP in patients with previously untreated DLBCL, with favorable response rates [41]. Importantly, with the staggered dosing, patients received glofitamab in the second cycle, the rates of CRS were markedly reduced compared to rates noted with the drug as a single agent in the R/R setting. Epcoritamab is being evaluated in combination with R-CHOP in the front line. Preliminary results from a phase 1/2 study showed an ORR and CRR of 100% and 90%, respectively, among 10 patients who had received six cycles of therapy [43]. Epcoritamab is being evaluated in combination with R-CHOP vs. R-CHOP alone in a phase 3 study for untreated DLBCL. Pending the results of these trials, the number of patients available to receive treatment with a BsAbs would increase exponentially. Evaluating BsAbs in front-line regimens in both DLBCL and FL is encouraging, as side effects and safety seem to be manageable, and there may be an improvement in therapeutic response and durability. However, future studies are needed to completely determine the role of BsAbs in front-line management.

BsAbs are also being combined with other novel/targeted agents in R/R DLBCL in multiple clinical trials, as previously mentioned. Mosunetuzumab plus polatuzumab vedotin was evaluated in R/R DLBCL in a phase 1/2 study, with an ORR and CRR of 62.2% and 50%, respectively, and a favorable side-effect profile [44]. Similarly, glofitamab has been combined with polatuzumab vedotin with encouraging response rates and safety [45]. In both situations, it appears that polatuzumab helped improve the efficacy of the BsAB, as well as reduced the frequency and severity of CRS. Epcoritamab with gemcitabine plus oxaliplatin was evaluated in a phase 1/2 study, with an ORR and CRR of 23/25 (92%) and 15/25 (60%), respectively, with zero grade 3 or higher CRS toxicities [46]. Similarly, epcoritamab was combined with rituximab, dexamethasone, cytarabine, and oxaliplatin or carboplatin (R-DHAX/C) in a phase 1/2 trial in transplant-eligible patients, with an ORR of 100% and CRR of 82%. CRS occurred in 8/27 (30%) patients, all of which were grade 2 or grade 3 [47]. Lastly, epcoritamab was combined with lenalidomide in r/r DLBCL in a phase 1/2 study, with a CRR of 14/26 (58%) and grade 3 CRS rates of 8% [48]. 

In FL, both epcoritamab and mosunetuzumab have been combined safely and effectively with lenalidomide. Phase 2 and 3 trials are ongoing for these combinations and could potentially supplement lenalidomide and rituximab as the preferred 2 L option.

Currently, there are novel BsABs that are being developed for NHL that have targets outside of CD20. The agent duvortuxizumab is a CD19 × CD3 that has been shown to have potent anti-tumor activity in preclinical B-cell lymphoma models [49]. Similarly, receptor tyrosine kinase-like orphan receptor 1 (ROR1) BsABs have been developed, and in vitro and in vivo studies suggest favorable antitumor activity, with clinical studies expected soon [50]. CD30 is another expected target for BsAB for use in NHL in both B- and T-cell malignancies [51]. Lastly, BsABs that harness NK cells, as opposed to T-cells, have also been developed with efficacy in early-phase human trials [52].

## 7. Conclusions

The development of BsAB has provided a well-tolerated and efficacious therapy in DLBCL, FL, MCL, and beyond. As described, BsAB will likely play an important role in the management of both front-line and R/R DLBCL and FL. These agents will likely broaden the clinical availability of novel immune based therapies in the community setting relative to CAR T-cell therapy given their “off the shelf” nature. Based on the current data, including response, duration of response, and toxicity, we included our current recommendations for sequencing bispecific antibodies in DLBCL Figure 1 and Figure 2. We recapitulated this for FL in Figure 3. As the data regarding BsAB mature and the agents move into earlier lines of therapy, how we re-use these agents in responding patients will likely become more important. The current FDA-approved agents of CD3XCD20 BsAB are listed in Table 1, and, looking forward, we listed some important upcoming studies in Table 2. However, agents that target CD19, ROR1, and CD30, as well as NK cells, are on the horizon. These developments foreshadow the excitement and bountiful potential of BsABs in NHL. Additionally, they continue to raise questions regarding how we should sequence and combine these novel therapeutics in future studies.

## Figures and Tables

**Figure 1 cancers-16-01724-f001:**
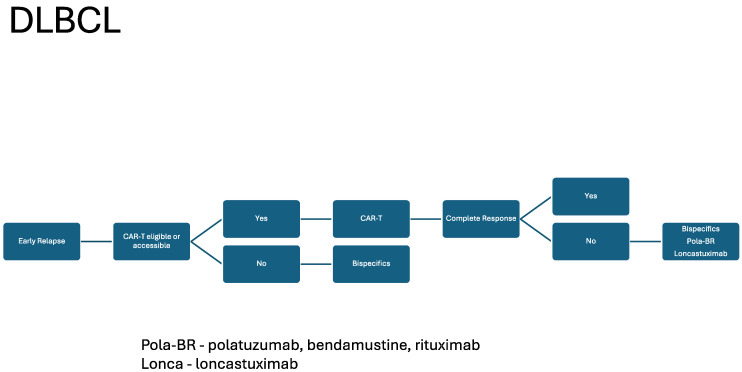
DLBCL: early relapse treatment schema.

**Figure 2 cancers-16-01724-f002:**
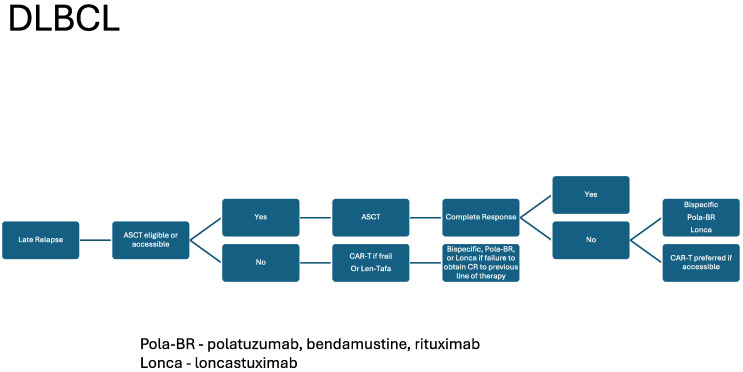
DLBCL: late relapse treatment schema.

**Figure 3 cancers-16-01724-f003:**
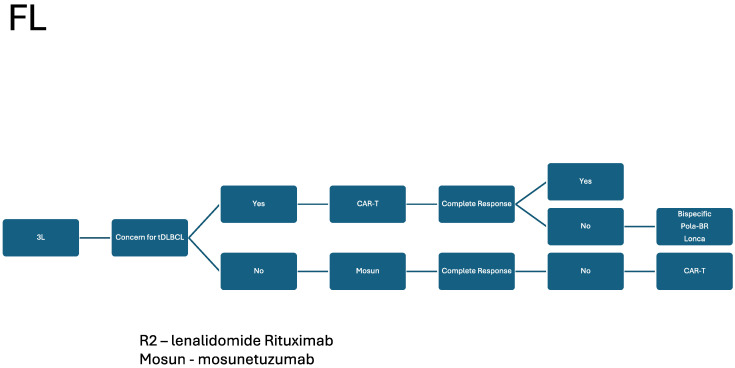
FL: treatment schema.

**Table 1 cancers-16-01724-t001:** Approved BsABs.

BsAB	Disease Type Approval	ORR	CRR	Median DOR	Post CAR T-Cell CRR
Epcoritamab [12]	DLBCL	63% (99/157)	39% (61/157)	15.6 months	34.4% (21/61)
Glofitamab [21]	DLBCL	52% (80/155)	39% (61/155)	18.4 months	32% (12/38)
Mosunetuzumab [13]	FL	80% (72/90)	60% (54/90)	22.8 months	NA

Abbreviations: BsAB = bispecific antibody; CRR = complete response rate; DOR = duration of response; ORR = overall response rate.

**Table 2 cancers-16-01724-t002:** BsAB combinations under investigation.

DLBCL	Clinicaltrials.gov ID	Line of Therapy
Glofitamab + Pola-R-CHP	NCT06050694, NCT03467373	Front
Glofitamab + RICE	NCT05364424	R/R
Epcoritamab + R-DHAOx	NCT06287398	R/R
Epcoritamab + lenalidomide	NCT05283720	R/R
Epcoritamab + ibrutinib	NCT05283720	R/R
Epcoritamab + Pola-R-CHP	NCT05283720	Front
Mosunetuzumab + Pola–lenalidomide	NCT06015880	R/R
FL	Clinicaltrials.gov ID	Line of Therapy
Mosunetuzumab + Pola	NCT05410418	Front
Mosunetuzumab + tazemetostat	NCT05994235	Front
Mosunetuzumab + Pola + Obin	NCT05169658	Front
Mosunetuzumab + lenalidomide	NCT06284122, NCT04792502	Front
Mosunetuzumab + lenalidomide	NCT04712097	R/R
Epcoritamab + lenalidomide	NCT06112847	Front
Epcoritamab + lenalidomide + rituximab	NCT06191744	Front
Odronextamab + R-CHOP	NCT06097364	Front
Odronextamab + lenalidomide	NCT06149286	R/R

Abbreviations: CHP = cyclophosphamide, doxorubicin, and prednisone; CHOP = cyclophosphamide, doxorubicin, vincristine, and prednisone; DHAOx = dexamethasone, cytarabine, and oxaliplatin; Obin = obinutuzumab; Pola = polatuzumab vedotin; R = rituximab; RICE = rituximab, ifosfamide, carboplatin, and etoposide.

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
