# Peer review of "Taking a BiTE out of Lymphoma: Bispecific Antibodies in B-Cell Non-Hodgkin Lymphoma"

_cancers, 2024, doi:10.3390/cancers16091724_

Round 1
Reviewer 1 Report
Comments and Suggestions for Authors
The paper is well written and summarize most recent data about BsAbs in DLBCL and FL. The authors critically discuss also recently approved therapies and their sequential use in relapsed/refratory DLBCL and FL. Upcoming BsAb combination with other drugs are also repoted as well as references to new developing BsAbs.
Just a minor question. Among BsAbs for DLBCL, the authors do not cite odronextamab which was studied in a DLBCL cohort of ELM2 trials with results comparable to those obtained for Glofitamab ed Epcoritamab. Do the author think that Odronextamb will have no role in future for DLBCL?
Author Response
Reviewer 1
Comments and Suggestions for Authors
The paper is well written and summarize most recent data about BsAbs in DLBCL and FL. The authors critically discuss also recently approved therapies and their sequential use in relapsed/refratory DLBCL and FL. Upcoming BsAb combination with other drugs are also repoted as well as references to new developing BsAbs.
Just a minor question. Among BsAbs for DLBCL, the authors do not cite odronextamab which was studied in a DLBCL cohort of ELM2 trials with results comparable to those obtained for Glofitamab ed Epcoritamab. Do the author think that Odronextamb will have no role in future for DLBCL?
Thanks for this question. Initially, we had focused on the two approved agents: glofitamab and epcoritamab. I have added a short update mentioning odronextamab, and cited the phase II ELM-2 as well, as this therapy may have a future in r/r DLBCL.
Reviewer 2 Report
Comments and Suggestions for Authors
In this manuscript, the Authors discuss novel bispecific antibodies (BsAbs), focussing on agents that have been approved for use in DLBCL and FL, and also give an overview of BsAbs that are currently in clinical trials in DLBCL and FL.
The review is easy to read and provides a concise overview of the key findings related to the use of BsAbs in DLBCL and FL. My only suggestion would be to add some information about the use of BsAbs in MCL considering the promising data from clinical trials in relapsed/refractory MCL.
Author Response
Reviewer 2
Comments and Suggestions for Authors
In this manuscript, the Authors discuss novel bispecific antibodies (BsAbs), focussing on agents that have been approved for use in DLBCL and FL, and also give an overview of BsAbs that are currently in clinical trials in DLBCL and FL.
The review is easy to read and provides a concise overview of the key findings related to the use of BsAbs in DLBCL and FL. My only suggestion would be to add some information about the use of BsAbs in MCL considering the promising data from clinical trials in relapsed/refractory MCL.
Thanks for the suggestion. We agree, a section on the use of BsABs in MCL would be worthwhile and informative. We have added a section on MCL that highlights the promising data in this disease type.